



# Properties of baroclinic Rossby waves in the North Atlantic from eddy-resolving simulations of ocean circulation

Sylvain Watelet[1,2], Jean-Marie Beckers[2], Jean-Marc Molines[3], and Charles Troupin[2]

[1]Observation Scientific Service, Royal Meteorological Institute, Brussels, Belgium [current affiliation]
[2]Department of Astrophysics, Geophysics and Oceanography, GeoHydrodynamics and Environment Research Unit, FOCUS Research Unit, University of Liège, Liège, Belgium
[3]Université Grenoble Alpes/CNRS/IRD/G-INP, IGE, Grenoble, France

**Correspondence:** Sylvain Watelet (swatelet@uliege.be)

**Abstract.** The teleconnections between the North Atlantic Oscillation (NAO) and the variability of the Gulf Stream (GS) were extensively studied these last years, often exhibiting time delays between both phenomena. These time lags, usually ranging between 0–2 years, are sometimes explained by the hypothesis of baroclinic Rossby waves generated by the NAO in the central North Atlantic (NA) and travelling westward before interacting with the GS. In this study, we use a numerical hindcast
at an eddy-resolving resolution (1/12°) from the DRAKKAR project to examine the occurrence and properties of such Rossby waves between 1970–2015, thus including a large pre-TOPEX/Poseidon period. Through the use of a two-dimensional Radon Transform (2D-RT) on Hovmöller diagrams of the Sea Surface Height (SSH), a methodology easily portable to other oceanic model outputs, we show evidence of baroclinic Rossby waves travelling at 39°N at a speed of $4.17\,\mathrm{cm\,s^{-1}}$. This is the first time Rossby waves are found that much north during such an extended period. These results are consistent with the time lags
observed between the NAO and the GS transport while the GS latitudinal shifts might obey additional processes.

## 1  Introduction

Although there is growing evidence of teleconnections between the North Atlantic Oscillation (NAO) and the variability of the Gulf Stream (GS) path and transport (e.g. Taylor and Stephens, 1998; Joyce et al., 2000; De Coëtlogon et al., 2006; Sasaki and
Schneider, 2011; Watelet et al., 2017), the physical causes are yet to be further explored in the light of the literature suggesting the time lags might be attributed to Rossby waves carrying the NAO signal from the central North Atlantic (NA) towards the GS region. Given the long time lags involved, establishing a cause and effect relationship between NAO and GS is fundamental to boost the predictability of the GS characteristics.

Rossby waves in the NA were first observed by Chelton and Schlax (1996) using 3-year time series from TOPEX/Poseidon
altimeters, before Cipollini et al. (1997) confirmed their presence around 34°N by combining Sea Surface Height (SSH) from TOPEX/Poseidon and Sea Surface Temperature (SST) from ERS-1 radiometer. They developed an objective detection tech-



nique based on Hovmöller (or longitude-time) diagrams (Hovmöller, 1949) of SSH or SST and Fast Fourier transforms applied on them, the method being extensively described in Cipollini et al. (2006). This technique allowed the detection of what are suggested to be the three first baroclinic modes of Rossby waves propagation, with estimated speeds of 2.7, 1.6 and $0.8\,\mathrm{km\,d^{-1}}$

(i.e. 3.1, 1.9 and $0.93\,\mathrm{cm\,s^{-1}}$) at 34°N and between 37-8°W. De Coëtlogon et al. (2006) later used similar Hovmöller diagrams based on transport outputs from an Oceanic General Circulation model (OGCM) to explain the 2-year time lag between the NAO and the GS transport, and found evidence for baroclinic Rossby waves travelling at speeds close to 3.5 and $2.5\,\mathrm{cm\,s^{-1}}$ at 27°N and 32°N, as previously computed by Chelton et al. (1998). They also suggested the fast (less than 1 month) response of the GS transport to the NAO could be due to faster barotropic Rossby waves. Osychny and Cornillon (2004) also used

TOPEX/Poseidon SSH data to show evidence of Rossby waves, in particular at 39°N in the NA, where they travel at estimated speeds comprised between $3\text{–}4\,\mathrm{cm\,s^{-1}}$. Finally, Lecointre et al. (2008) showed a high-resolution (1/6°) numerical model can generate Rossby waves at speeds similar to those detected from altimetry.

The present study aims at detecting Rossby waves from a state-of-the art eddy-resolving model. The region of interest is located around the latitude of the average GS path (∼39°N). At these relatively high latitudes, Rossby waves detection

is still challenging, and only a few studies can be found in the literature (e.g. Osychny and Cornillon, 2004). The use of a numerical model instead of satellite data has the advantage of being independent of the satellite constellation as well as allowing future subsurface exploration of these waves. Besides, the simulations used here cover a longer time span including the pre–TOPEX/Poseidon era (i.e. before 1992), which fosters the detection of such slow signals.

## 2    Data and methods

We have used the outputs of a global hindcast performed in the frame of the DRAKKAR project (Barnier et al., 2006, 2007) at the eddy-resolving resolution of 1/12° with the ORCA12 configuration described here (https://github.com/meom-configurations/ ORCA12.L46-MJM189). This hindcast is based on the NEMO ocean / sea ice GCM numerical code (Madec and NEMO-team, 2016). In the past, similar simulations based on OPA 8.1 were performed during the CLIPPER experiments and compared to Eddy Kinetic Energy (EKE) satellite observations by Penduff et al. (2004) whereby concluded to a reasonable agreement.

In addition, such simulations showed a good agreement with altimetry when considering Rossby waves phase occurence and speeds (Lecointre et al., 2008), which is a prerequisite for our study. Both studies were however performed with a 1/6° ORCA grid, which is only eddy-permitting at the latitudes of interest. Besides, Barnier et al. (2006) also showed the introduction of partial steps topography can significantly improve mean flow and EKE representation, leading to simulations at 1/4° (NEMO) performing as well or even better than above-mentioned 1/6° simulations (OPA 8.1). Using this numerical development, Pen-

duff et al. (2010) and then Sérazin et al. (2015) showed how increasing the resolution from 1–2° to 1/4° or 1/12° allows a better agreement between altimetry and modeled SSH. We therefore expect a quantitative improvement in the Rossby waves detection from the use of an enhanced resolution.

This simulation covers the period 1958–2015 and was forced by the interannual DRAKKAR Forcing Set DFS5.2 (based on a rescaling of ERA-Interim ECMWF atmospheric re-analysis) (Dussin et al., 2018). The relevant parametrisations for this study





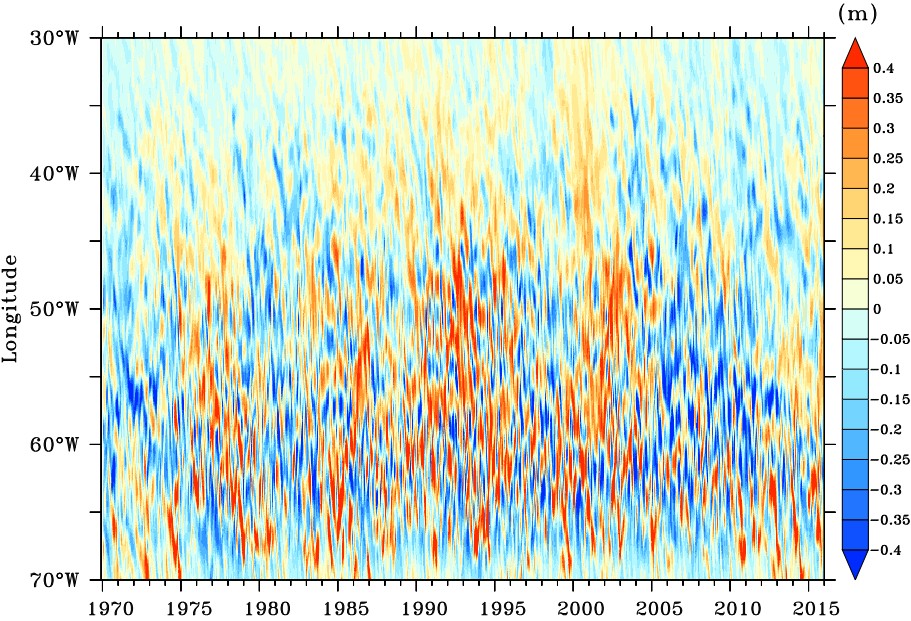

**Figure 1.** Hovmöller diagram of detrended SSH anomalies at 39°N between 30–70°W and 1970–2015, showing Rossby wave-like patterns. A running average on 30 days and 28 km is applied.

are: linear filtered free surface technique to avoid fast barotropic waves, biharmonic horizontal lateral viscosity, and isopycnal lateral diffusivity. Vertical mixing is controlled by the TKE scheme (Blanke and Delecluse, 1993; Madec et al., 1998).

The basic data set consists of the SSH fields (5–day average) model output, for the period 1970–2015, extracted from the global simulation on the NA (0–65°N). These fields are available at https://doi.org/10.5281/zenodo.3968801 between 38 and 40°N (Molines, 2020). The chosen period allows for a 13–year spin-up of the simulation. We decided to focus on the latitude

39°N which was considered representative of the average GS path from our SST fields in Watelet et al. (2017). Here, indeed, we only examine the zonal component of the Rossby waves travelling across the NA. Regarding longitudes, we chose to analyse propagating signals from 30 to 70°W, i.e. between the approximate center of the NA at 39°N where we can expect the NAO signal to be carried to the ocean through wind forcing and the western limit of the GS region as defined in Watelet et al. (2017).

These SSH fields, originally provided at a horizontal resolution of 1/12° (ORCA12 grid), were interpolated bilinearly to

a constant 1/12.5° or 0.08° grid in order to avoid truncation issues with repeating decimals. From there, we removed the climatological annual cycle as well as the 1970–2015 trend. The data set is available at https://doi.org/10.5281/zenodo.3968885 (Watelet, 2020). Figure 1 shows these detrended SSH anomalies as an Hovmöller (or lon-time) diagram. On this diagram, Rossby wave-like patterns are already visible as the lines of similar SSH with a slope tilted to the left and close to the vertical. Looking at a much shorter time scale (see Figure 2), between 1985–1988, it is even possible to draw subjective slopes (dotted

lines) of these waves, allowing a rough estimate of their speed.





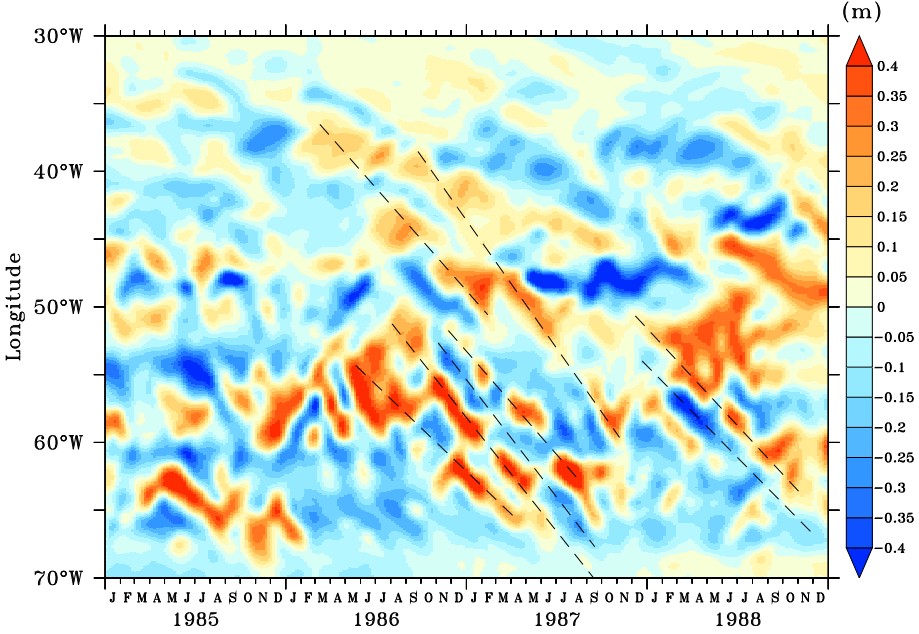

**Figure 2.** Hovmöller diagram (longitude–time) of detrended SSH anomalies at 39°N between 30–70°W and 1985–1988. A running average on 30 days and 28 km is applied. Black dotted lines represent possible examples of Rossby waves.

Detrended SSH anomalies between 70-30°W and 37-43°N averaged on 30 days also make it possible to detect features propagating westward, especially in the eastern part of the domain. West of 50°W, these waves are more difficult to discern, probably because this relatively weak and varying signal interacts with the strong flow of the GS and its meanders (see the animated GIF in Supplementary Material).

With the aim of avoiding subjective biases, we then followed the method explained in detail in Cipollini et al. (2006). At first, the SSH fields were averaged with moving 30-day and 28-km boxcar windows in order to avoid short space and time scales noise, as shown in Figure 1. Then, a two-dimensional Radon Transform (2D-RT) as defined by Deans (1983) and Challenor et al. (2001) was applied to this Hovmöller diagram. The idea of this 2D-RT is to perform a rotation of the $\frac{time}{dt} - \frac{lon}{dlon}$ (x,y) dimensionless coordinates (dt = 5 days, dlon = 0.08°) by an angle $\varphi$ before summing the SSH for each x' (x rotated) on all y'

(y rotated). Thus, each (x',y') is associated to (x,y) as follows:

$$x = x^{'}\cos(\varphi) - y^{'}\sin(\varphi) \tag{1}$$

$$y = x^{'}\sin(\varphi) + y^{'}\cos(\varphi) \tag{2}$$






allowing the calculation of the 2D-RT from the SSH in Figure 1:

$$RT(\varphi, x^{'}) = \int_{y^{'}} SSH(x,y) dy^{'} \tag{3}$$

The 2D-RT is here computed for a set of angles $\varphi$ ranging from 0 to 90° by steps of 1°. For each angle, we then computed the energy of the 2D-RT:

$$RTE(\varphi) = \int_{x^{'}} (RT(\varphi, x^{'}))^2 dx^{'} \tag{4}$$

The angle for which the 2D-RT energy is maximum is logically the one for which the maximum energy propagates westward.

## 3   Properties of the Rossby waves

Figure 3 shows the 2D-RT values for each time (x') and angle $\varphi$. The energy of this 2D-RT is shown in Figure 4, exhibiting a maximum when using an angle $\varphi$ of 21°. This angle is consistent with the Rossby waves visually detected on Figures 1 and 2 (where an angle of 45° is defined as 0.08° in 5 days). Its speed can now be objectively estimated at $4.17\,\mathrm{cm\,s^{-1}}$.

In order to compare this speed estimate with theory, we need to compute the Rossby wavenumber. Considering the RT at $\varphi = 21°$ (Figure 5), the first step in identifying this wavenumber is to perform a Fast Fourier Transform (FFT) on this 2D-RT, and then look for the leading period present in the signal. The first and last 5 years of the time series of the 2D-RT were removed beforehand to avoid edge effect. Figure 6 shows the power spectrum of this FFT for various periods between 0 and 2 years. The highest peak corresponds to a period of 269 days, while two other peaks can be seen at 190 and 467 days. The highest
peak represents the first baroclinic Rossby wave and the following peak might correspond to contamination from higher order Rossby waves, although Maharaj et al. (2004) showed this last question remains open. In order to check the sensitivity of these periods against the angle of the RT, we carried out the same procedure for both RT angles of 20 and 22°. The resulting power spectra in Figures A1 and A2 confirm the highest peak around 269 days with virtually identical periods for the 3 main peaks.
     From this leading period, the spatial wavelength is computed by projecting the period on the longitude axis of the Hovmöller
diagram in Figure 1. This yields a wavelength of 13.37° corresponding to 1066 km at this latitude. Hence, the wavenumber is estimated at $5.878 \times 10^{-6}\,\mathrm{m^{-1}}$. The beta parameter $\beta_0$ is computed as follows:

$$\beta_0 = 2\frac{\Omega}{a}\cos(\alpha_0) \tag{5}$$

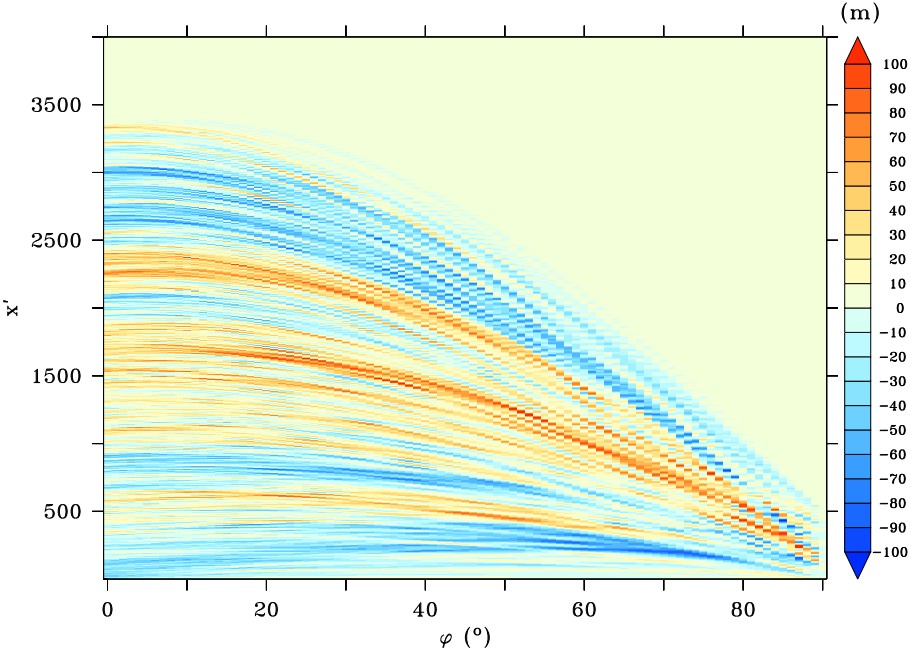

**Figure 3.** 2D-RT as defined in Eq. 3 computed on detrended SSH anomalies shown in Fig. 1, after applying 30-day and 28-km boxcar smoothings.

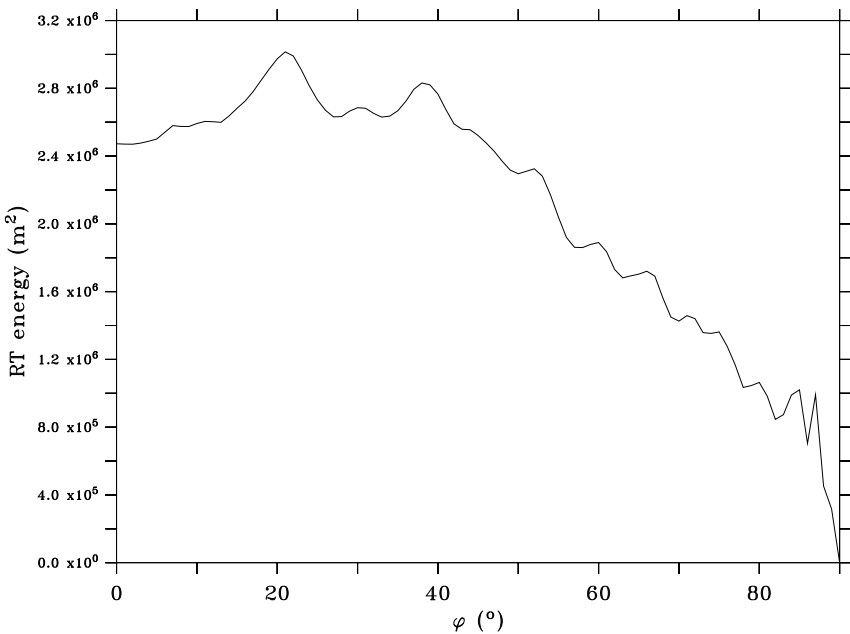

**Figure 4.** 2D-RT energy as defined in Eq. 4 computed on detrended SSH anomalies shown in Fig. 1, after applying 30-day and 28-km boxcar smoothings.


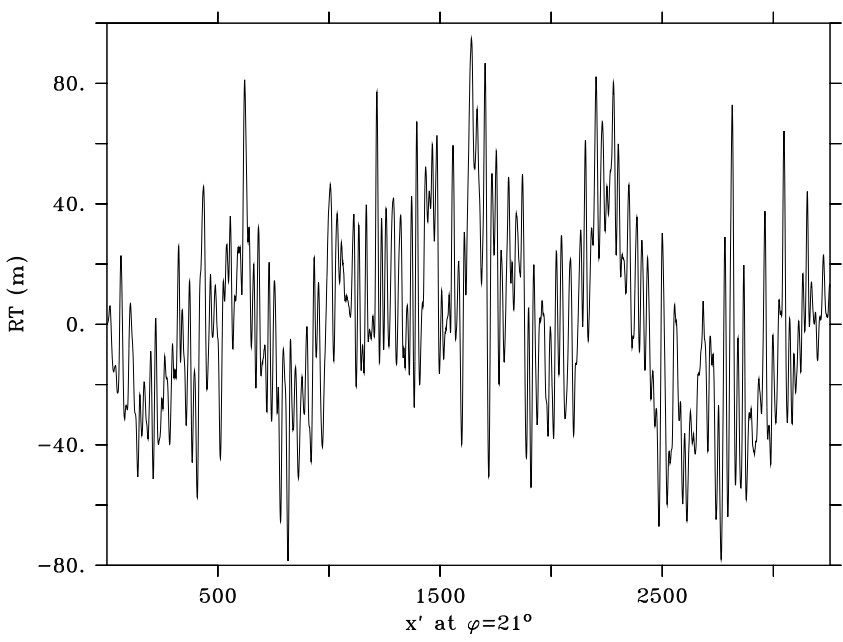

**Figure 5.** RT at $\varphi = 21°$ extracted from Fig. 3.

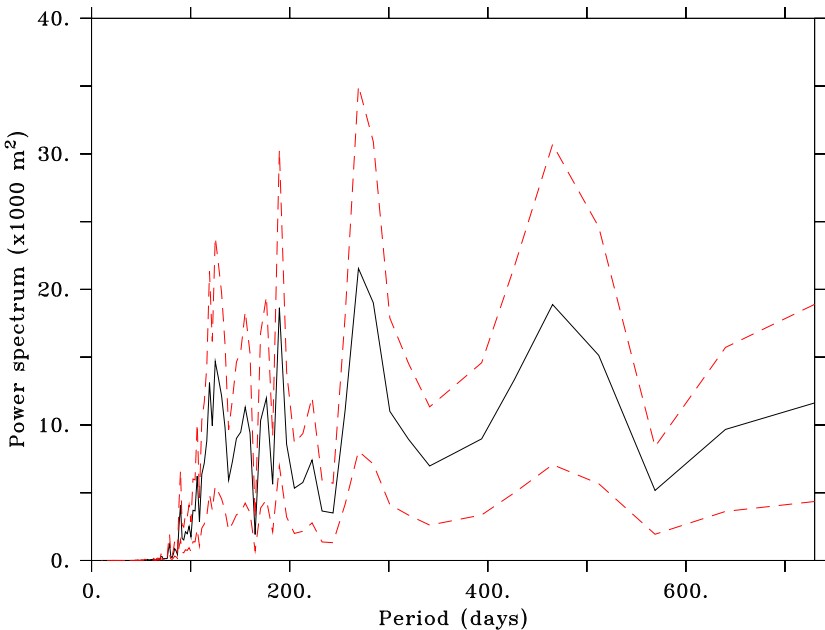

**Figure 6.** Power spectrum of the FFT applied to the RT at $\varphi = 21°$ for periods ranging between 0–2 years. Between both dashed red lines, an uncertainty of two standard deviations is shown.





where $\Omega$ is the angular speed of the Earth's rotation ($7.29 \times 10^{-5}$ rad/s), $a$ is the Earth's radius (6371 km) and $\alpha_0$ the latitude of the wave (39°N). In this case, $\beta_0$ equals $1.77 \times 10^{-11}\,\mathrm{m^{-1}\,s^{-1}}$.

The first baroclinic Rossby radius of deformation R is then estimated following the atlas published by Chelton et al. (1998). In the GS region, R is close to 30 km which is thus used in the calculation hereafter.

For this set of parameters, one can show that the group speed computed from our Radon analysis should be similar to the corresponding phase speed. Indeed, the dispersion relation of the Rossby waves reads:

$$\omega = -\beta_0 R^2 \frac{k_x}{1 + R^2(k_x^2 + k_y^2)} \tag{6}$$

and the zonal phase speed that follows:

$$c_x = \frac{\omega}{k_x} = \frac{-\beta_0 R^2}{1 + R^2 k_x^2} \tag{7}$$

where we used $k_y = 0$. From these equations, the group speed is defined as:

$$c_{xg} = \frac{\partial \omega}{\partial k} = -\beta_0 R^2 \frac{1 - 2R^2 k_x^2}{1 + R^2 k_x^2} \tag{8}$$

which is close to $c_x$ provided we use typical values of $\sim 10^{-6}\,\mathrm{m^{-1}}$ for $k_x$ and $\sim 10^4$ m for $R$, since $2R^2 k_x^2$ is then negligible when compared to 1.

We can thus consider the speed computed from our Radon analysis as a phase speed, and compare it with the theory using the parameters above and the equation 7. This yields a theoretical phase speed of $1.55\,\mathrm{cm\,s^{-1}}$, 2.7 times weaker than our empirical estimate.

Such a discrepancy between theory and practical estimates of baroclinic Rossby wave speeds have been encountered by many authors (e.g. Chelton and Schlax, 1996; Osychny and Cornillon, 2004; Maharaj et al., 2004). At 39°N, Osychny and Cornillon (2004) found a factor of discrepancy slightly larger than ours. The standard theory was accordingly adjusted by Killworth et al. (1997) and updated by Killworth and Blundell (2003, 2004, 2005) to account for the effects of the baroclinic background mean flow and topographic gradients, the former being generally dominant according to Maharaj et al. (2007). As reported by Killworth et al. (1997), the discrepancy factor between observed speeds and those predicted by standard theory



reaches 2 poleward of 30°N. Besides, their estimated maximum ratio between standard and extended theory reaches 3 at 39°N, which is consistent with our estimate of 2.7.

## 4    Summary and discussion

This study shows that it is possible to detect Rossby waves at latitudes compatible with the average position of the GS while using long time ranges covering pre-altimetry era, provided sufficiently resolved numerical simulations are used. The methodology described here was applied to 1/12° SSH outputs from the DRAKKAR project, but it can easily be adapted for other models or variables. In particular, it would be instructive to examine the sensitivity of the Rossby waves detection to an even finer spatial resolution.

Following the state of the art, the time lag between a specific NAO phase (+ or –) and its consequence on the GS path and transport still remains a question on which there is no perfect consensus at this stage. As a matter of fact, the bulk of estimates ranges between 0 and 2 years considering both GS path and transport (Watelet et al., 2017).

Considering the NAO signal transfers momentum through the wind stress to the central part of the NA (Visbeck et al., 1998), with a maximum impact on the SSH in this area (Esselborn and Eden, 2001), we use hereafter the longitude 30°W as a reference for the perturbation initiating a westward SSH Rossby wave. In order to compare with GS indices from Watelet et al. (2017) established by considering EOF's computed on 81 equally spaced longitudes between 70°W and 50°W, we use the longitude 60°W as representative of the place where the incoming Rossby wave would impact the characteristics of the GS. These assumptions leads to 30° to be travelled westward at the latitude 39°N. Using our phase speed estimate of $4.17\,\mathrm{cm\,s^{-1}}$ or $15.22\,^\circ\mathrm{yr^{-1}}$ yields a travel time of 1.97 years (∼24 months), which is consistent with usual NAO–GS delays.

Still, the major part of GS indices are computed on a yearly basis, for both position and transport, which leaves room for deeper investigation. We thus used the monthly GS Delta index (GSD, proxy for its transport) from Watelet et al. (2017), already averaged on running 12 months, and used their monthly GS North Wall index (GSNW, representative of its position) that we averaged the same way. This allows the computation of correlations with NAO using monthly time lags, as shown in Figure 7. The NAO indices we use in this Section are the Hurrell annual or monthly NAO index, both based on an EOF analysis of the sea level pressure over the NA (see National Center for Atmospheric Research Staff, 2015; Hurrell, 1995; Hurrell et al., 2003; Hurrell and Deser, 2010; Trenberth and Hurrell, 1999). Both are available online (at https://climatedataguide.ucar.edu/climate-data/hurrell-north-atlantic-oscillation-nao-index-pc-based).

The correlations are shown for the period 1960-2014, chosen as the longest reliable period for the GSD index. The NAO–GSD correlation peaks at time lags of 1 and 29 months, possibly indicating the influence of a fast barotropic Rossby wave followed by its slower baroclinic counterpart, in accordance with De Coëtlogon et al. (2006). Although this last delay is consistent with the baroclinic waves we detected, the interpretation of the NAO–GSNW correlations remains unclear, with significant correlations between 3 and 14 months, peaking at 7 months. Nevertheless, the exact value of this time lag is still discussed and depends on methods, periods and data used (e.g. Taylor and Stephens, 1998; Joyce et al., 2000; Watelet et al., 2017).


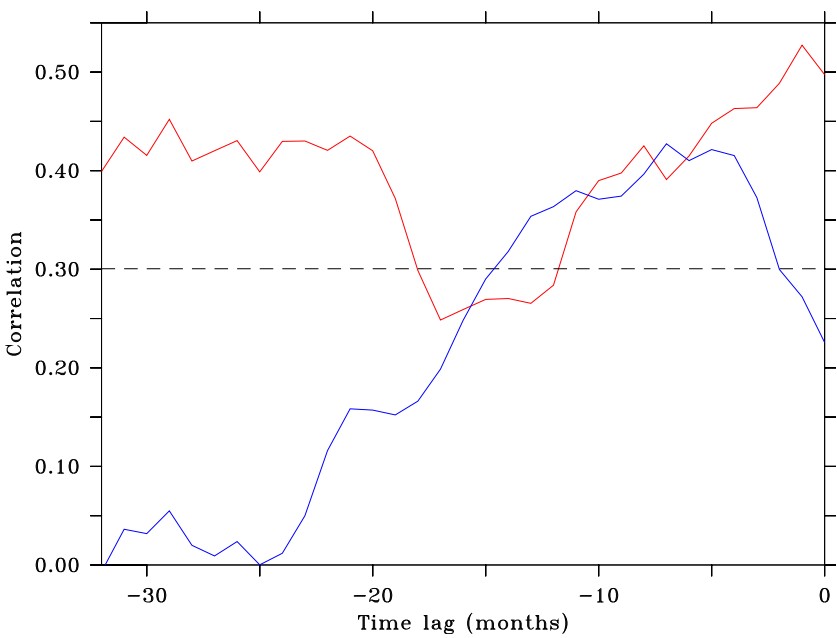

**Figure 7.** Correlations between monthly NAO–GSD (red) and NAO–GSNW (blue) for time lags ranging between 0 and 32 months. The black dotted line shows the significance threshold at a 95% confidence level.

Apart from the influence of westward Rossby waves, another mechanism was proposed to account for the GSNW shifts: the fluctuations in the southward flow of Labrador Sea water might impact the GS path (Rossby, 1999). Hameed and Piontkovski (2004) even suggested this last mechanism to be preponderant against the influence of the NAO through Rossby waves, by showing the Icelandic Low has a dominant influence on the GSNW while the Labrador Sea wind stress is mainly connected to
the same Icelandic Low. While outside the scope of this paper, this second mechanism might explain our difficulties to find a straightforward link between the NAO and the GSNW.

Finally, in order to further explore the link between the NAO and the baroclinic Rossby waves, we compare the monthly NAO index to the RT at $\varphi = 21°$ between 1970–2012. We assigned time values to the RT by considering the approximate moment at which the wave is generated by the NAO signal, i.e. we attributed the time corresponding to 30°W for a particular
SSH wave. This explains why we had to somewhat shorten the original 1970–2015 period. Then, we smoothed both signals by using a 9-month running average, and normalised them to get Figure 8. This smoothing length is chosen as the estimated period of Rossby waves. The positive correlations between unsmoothed NAO and RT (not shown) and between both smoothed indices are not significant. Nevertheless, there are visual similarities between smoothed NAO and SSH waves, especially looking at multi-year time scales. Besides, we show in Figure 9 that the direct correlations between NAO and SSH between 50–30°W
and 1970–2015 exhibit small peaks at time lags increasing westward (red line), consistently with the hypothesis of the NAO generating a Rossby wave around 30°W. The linear regression (black dashed line) between these peaks is significant and reveals a Rossby wave speed of $12.67\,°\,\mathrm{yr}^{-1}$ or $3.47\,\mathrm{cm\,s}^{-1}$, which is close to our previous estimate (black solid line). Despite this

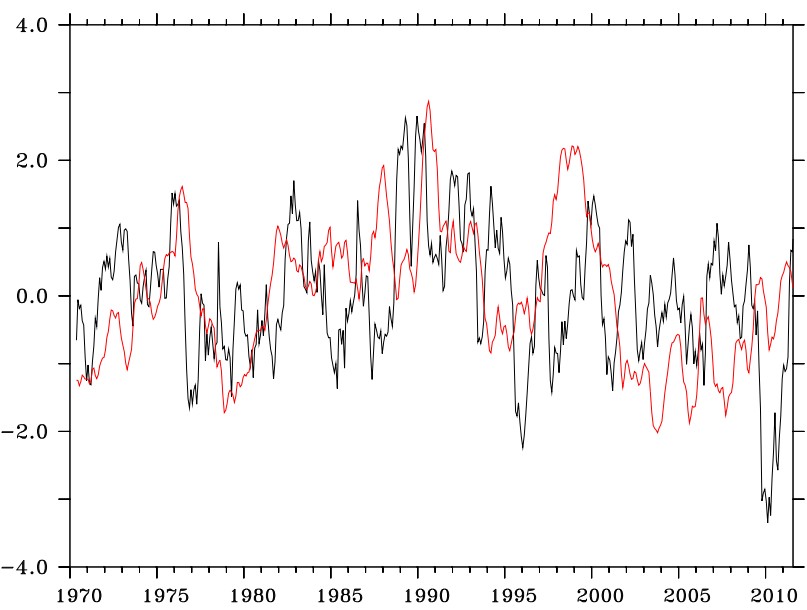

**Figure 8.** Normalised NAO index (black) and RT at $\varphi = 21°$ (red) between 1970–2012. A 9-month smoothing was applied on both indices.

body of corroborating evidence, we believe there is room here for further research on the physical processes linking NAO and Rossby waves. In particular, the connection between a specific NAO phase and the generation of oceanic Rossby waves has

been rather neglected so far and thus requires more numerical experiments, such as sensitivity tests of Rossby waves intensity to various wind stress forcings.

*Code and data availability.* The code used to perform the DRAKKAR simulations is available at https://doi.org/10.5281/zenodo.3968307. The data sets used in this study are available at https://doi.org/10.5281/zenodo.3968801 and https://doi.org/10.5281/zenodo.3968885.





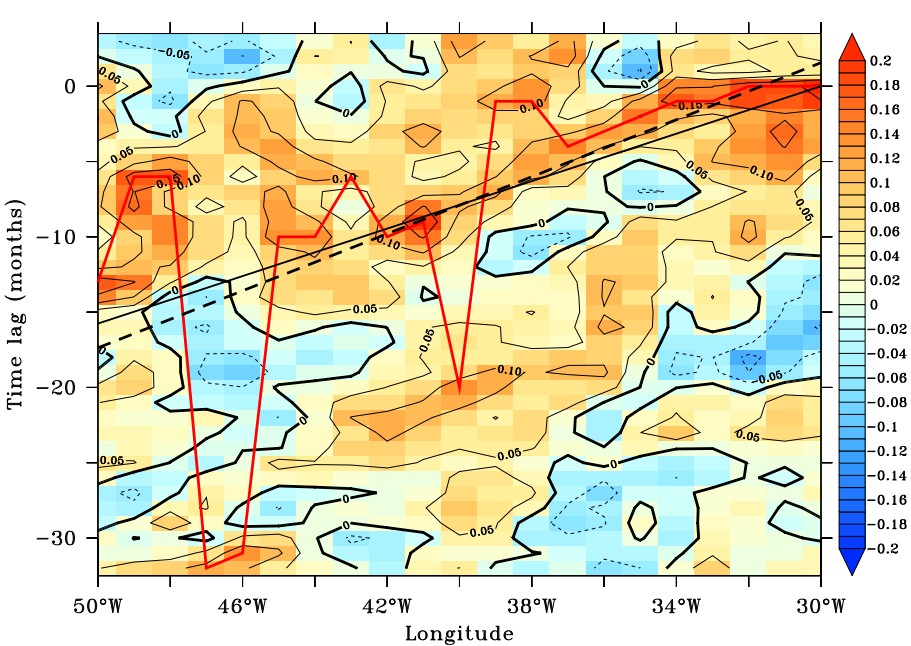

**Figure 9.** Correlations between monthly NAO index and DRAKKAR SSH between 1970–2015 for time lags ranging from -32 to +3 months (negative lags means NAO precedes SSH) and for longitudes between 50–30°W. The red line shows the maximum correlation for each longitude, while the black dashed line is its linear regression with a slope of $0.947 \, \mathrm{mth} \, {}^{\circ-1}$ corresponding to a propagation speed of $12.67 \, {}^{\circ} \mathrm{yr}^{-1}$. The black solid line corresponds to our estimated Rossby wave speed from RT (see previous Sections).





## Appendix A: Power spectra

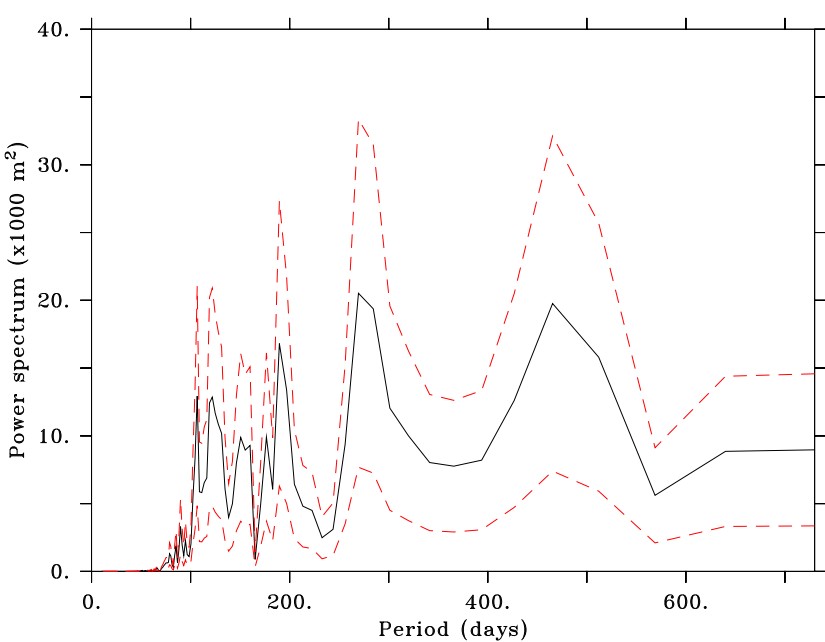

**Figure A1.** Power spectrum of the FFT applied to the RT at $\varphi = 20°$ for periods ranging between 0–2 years. Between both dashed red lines, an uncertainty of two standard deviations is shown.



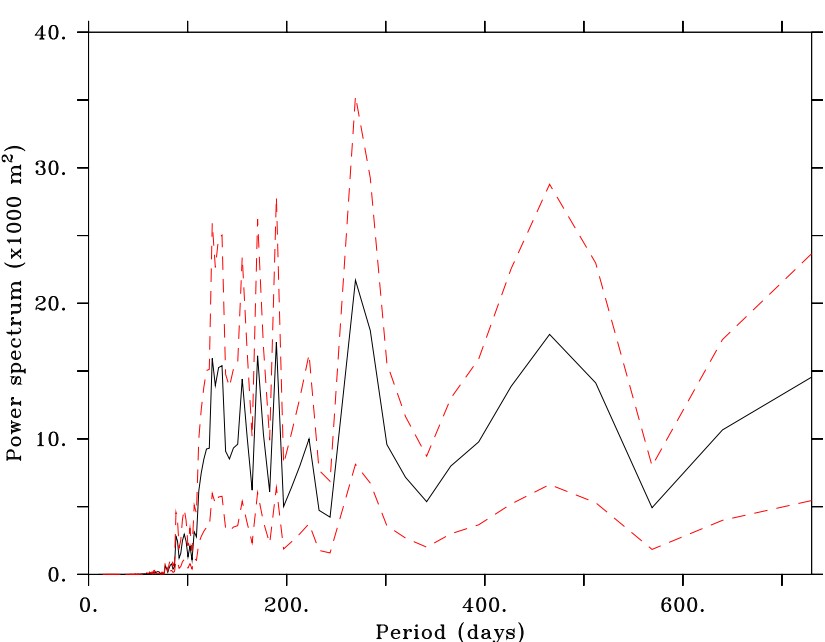

**Figure A2.** Power spectrum of the FFT applied to the RT at $\varphi = 22°$ for periods ranging between 0–2 years. Between both dashed red lines, an uncertainty of two standard deviations is shown.





*Author contributions.* Sylvain Watelet conducted the research and prepared the manuscript with contributions from all co-authors. Jean-Marie Beckers contributed in designing the research. Jean-Marc Molines prepared the DRAKKAR data. Charles Troupin helped proof-reading the manuscript.

*Competing interests.* The authors declare that they have no conflict of interest.

*Acknowledgements.* We would like to thank the DRAKKAR team for their availability in preparing and sharing the numerical simulations
we used in this research, and particularly Bernard Barnier for his valuable comments on those. This is a FOCUS publication.





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
