# Peer review of "Properties of baroclinic Rossby waves in the North Atlantic from eddy-resolving simulations of ocean circulation"

_Ocean Science, 2020_

## Referee Comment (RC1) · Anonymous Referee #1 · 29 Sep 2020

The authors apply two dimensional signal processing to time/longitude model output of ssh variability. They identify westward propagation of features with a phase speed of 4.17 cm/s. The Rossby wave dispersion relation, taking a reasonable value for the baroclinic deformation radius, predicts a phase speed of 1.55 cm/s. The difference is consistent with previous obervations that Rossby waves tend to propagate faster than free wave theory predicts. The authors then go on to try to relate the NAO to the Gulf Stream position and the excitation of Rossby waves.

The application of the 2DRT seems useful and valid. The problem of how mid-latitude Rossby waves are excited and influence the Gulf Stream is important. However, while

[Figure]

I do not find any technical problems with this analysis, I also do not find it particularly enlightening. There just isn't enough new here, my recommendation is to reject the paper.

The authors cite several previous works that have identified Rossby waves north of 35N, so the point in the Abstract that this is the first time Rossby waves have been found that far north for such an extended period is not that novel.

There is little mention of the influence of the Gulf Stream on the wave propagation. Even though the meridional wavelength is likely large compared to the Gulf Stream width, the authors apply their analysis right in the latitude band of the strong eastward flow. What happens if the analysis is applied a little further to the south?

Analysis demonstrating a lagged correlation between GSNW and NAO is not new and the present analysis, while consistent with previous work, does not add much to the paper.

It was unclear to me how RT is used to generate a time series. Please expand on the discussion around lines 185.

The results in Figs 8 and 9 are interesting but unfortunately the lack of statistical significance makes this a less than compelling argument.

---

## Referee Comment (RC2) · Anonymous Referee #2 · 11 Nov 2020

The authors make an attempt at identifying and isolating westward propagating baroclinic Rossby wave signals in the North Atlantic in a model simulation forced by observed atmospheric forcing. While the premise is not unjustified, I found the results to be entirely unconvincing.

For instance, Figure 1 only visually reveals weak signatures of westward propagation and the lines in Figure 2 (smoothed version) are not at all convincing. A wave should be a continuous, linear local peak in a Hovmoller plot. But the lines are drawn over short periods of one-signed anomalies, connected by opposite signed anomalies. That is not evidence of continuous propagation. Perhaps a better interpretation is that waves are

generated over short distances, then damped or interrupted by other processes.

Figure 4 shows a relatively flat 2D-RT phase spectrum for angles less than 45 deg. Is there any evidence the purported peak at 21 deg is actually statistically significant over that background noise? Chelton recanted his concept of fast internal Rossby waves in the satellite observations by instead re-casting the preponderance of westward propagating signals as evidence of nonlinear eddies rather than waves. Only the largest scales seemed to be wavelike in behavior, but they were difficult to dissociate from the energetic mesoscale that dominated their analysis (Chelton et al., 2011). The authors in the present study give no indication that they are aware of or concerned about this updated framework.

The SSH signals include components from multiple vertical modes, which can obscure the signal they are after. Since they have a model, it would be more far more interesting to analyze the full model fields rather than just looking at SSH. Separate the flows into vertical modes, including the barotropic and 1st baroclinic, and maybe even the 2nd baroclinic, so that you can be more unambiguous in following only one internal mode, presumably the 1st baroclinic.

It's not just the NAO forcing that drives the waves. It is the zonally and temporally integrated upstream effects of the wind-stress curl for all time scales at each longitude that matters locally in the oceanic response. Hence, showing lagged correlations of indices in Figure 7 is not particularly interesting or illuminating.

Finally, Figure 8 looks like two (essentially) uncorrelated red-noise time series.

Minor points:

1) All the figure captions should indicate that "model SSH" is being analyzed and plotted.

2) Throughout the text, many of the numerical estimates indicate "four significant digits" accuracy. Really?

Reference: Chelton, et al., 2011: Global observations of nonlinear mesoscale eddies. Prog. Oceanogr., 91, 167-216.

---

## Author Comment (AC1) · 17 Dec 2020

**Reply to reviewer #1**

**We would like to thank the reviewer #1 for the time dedicated for the review of our work, and for the several ideas that allowed to improve the clarity of the manuscript. Our replies are in bold.**

The authors apply two dimensional signal processing to time/longitude model outputof ssh variability. They identify westward propagation of features with a phase speedof 4.17 cm/s. The Rossby wave dispersion relation, taking a reasonable value for thebaroclinic deformation radius, predicts a phase speed of 1.55 cm/s. The difference is consistent with previous obervations that Rossby waves tend to propagate faster than free wave theory predicts. The authors then go on to try to relate the NAO to the Gulf Stream position and the excitation of Rossby waves.

The application of the 2DRT seems useful and valid. The problem of how mid-latitude Rossby waves are excited and influence the Gulf Stream is important. However, while I do not find any technical problems with this analysis, I also do not find it particularly enlightening. There just isn't enough new here, my recommendation is to reject the paper.

The authors cite several previous works that have identified Rossby waves north of 35N, so the point in the Abstract that this is the first time Rossby waves have been found that far north for such an extended period is not that novel.

**Regarding the previous works identifying Rossby waves north of 35°N, we cited Osychny and Cornillon (2004) and Lecointre et al. (2008). Lecointre et al. (2008) used a numerical simulation at a spatial resolution of 1/6° between 1993-2000. Osychny and Cornillon (2004) used a TOPEX/Poseidon product at a spatial resolution of 1° between 1992-1998.**

**In this study, we use an eddy-resolving numerical simulation at a spatial resolution of 1/12° between 1970-2015. Therefore, we definitely use a much longer period (46 years against 6 to 8 years) and a higher spatial resolution (1/12° against 1/6° to 1°) than the two previous studies to detect Rossby waves that far north.**

**We suggest to rephrase "this is the first time Rossby waves have been found that far north for such an extended period" into "this study extends the period over which Rossby waves have been found that far north to a much longer period, which reinforces the findings of previous works".**

**In addition, we fully described in Section 2 the advantages of working with a higher spatial resolution to better detect the Rossby waves.**

There is little mention of the influence of the Gulf Stream on the wave propagation. Even though the meridional wavelength is likely large compared to the Gulf Stream width, the authors apply their analysis right in the latitude band of the strong eastwardflow. What happens if the analysis is applied a little further to the south?

**In this study, we focus on the delayed impact of the NAO on the Gulf Stream. Applying such an analysis to a latitude band that would not correspond to the Gulf Stream latitude is thus outside the scope of the paper.**

**We added a corresponding comment in the paper "Since this study focus on the delayed impact of the NAO on the GS, other latitude bands are considered outside the scope of this paper."**

Analysis demonstrating a lagged correlation between GSNW and NAO is not new and the present analysis, while consistent with previous work, does not add much to the paper.

**The correlations we present here between the NAO and the GSNW are simply recomputed on a monthly basis from the indices of Watelet et al. (2017). Looking at delays at a resolution of 1 month instead of 1 year is relevant in order to allow a comparison with the Rossby wave speeds. The main goal of this paper is not to demonstrate that lagged correlations between NAO and GSNW exist. The purpose of our study is to detect Rossby waves at the latitude of the GS, compute the speed of these waves, and analyse the consistency between these waves and the lagged correlations that were found previously. The fact that we refined the NAO-GSNW correlations to a monthly basis is thus a minor part of the paper, but still a necessary improvement to reach our main goal.**

**In addition, the exact time lag between NAO and GSNW is still discussed, as stated in Section 4. There is thus room for further research there as well.**

**In order to improve the clarity of the manuscript, we added the following comment: "Looking at delays at a resolution of 1 month instead of 1 year is necessary in order to allow an accurate comparison with the Rossby wave speeds."**

It was unclear to me how RT is used to generate a time series. Please expand on the discussion around lines 185.

**We added the following sentence to the manuscript: "In other words, we projected the axis x' on the original time axis x to get a time series from the RT at phi=21°." For clarity, we also added: "As a reminder, the RT computation is based on the sum of the SSH of a Hovmöller diagram along the spatial axis progressively tilted to the left as the angle phi increases."**

The results in Figs 8 and 9 are interesting but unfortunately the lack of statistical significance makes this a less than compelling argument.

**We agree that the Fig 8 and 9 do not show statistically significant correlations, which is described in the manuscript. We also think, similarly to the reviewer #1, that these figures are nevertheless interesting. These weak correlations are an argument to encourage further research in the physical processes linking NAO and Rossby waves.**

**We adapted the text by adding "While the correlations are not significant, the figures are nevertheless interesting. These weak correlations are an argument to encourage further research in the physical processes linking NAO and Rossby waves."**

---

## Author Comment (AC2) · 17 Dec 2020

**Reply to the reviewer #2**

**We would like to thank the Reviewer #2 for the time dedicated to the review of our work, and for the several ideas that allowed to improve the manuscript. Our answers are in bold.**

The authors make an attempt at identifying and isolating westward propagating baroclinic Rossby wave signals in the North Atlantic in a model simulation forced by ob-served atmospheric forcing. While the premise is not unjustified, I found the results to be entirely unconvincing.

For instance, Figure 1 only visually reveals weak signatures of westward propagation and the lines in Figure 2 (smoothed version) are not at all convincing. A wave should be a continuous, linear local peak in a Hovmoller plot. But the lines are drawn over short periods of one-signed anomalies, connected by opposite signed anomalies. That is not evidence of continuous propagation. Perhaps a better interpretation is that waves are generated over short distances, then damped or interrupted by other processes.

**Of course the reviewer #2 is right that the Hovmöller plot on Figure 1 does not show pure wave propagation, but the westard propagation can already be observed, which is why we illustrate it by lines corresponding to possible Rossby waves on Figure 2. As we stated in the text, this can only be subjective since observed with the human eye and we performed the 2D Radon transform covering the whole range of longitudes of the Figure 1 to get the dominant propagation signal. The Rossby waves speed that we consider objective is estimated only after this analysis. Besides, other than the spatial and temporal smoothing, we did not use any filtering method, which explains why the Rossby waves signal is not visually perfect on Fig 1 and 2. This method actually enhances our confidence in the presence of Rossby waves since we detected them despite not using many subjective filters.**

**In order to clarify the text, we have added the following sentences in the manuscript: "Other than these spatial and temporal smoothing, we did not use any filtering method, which explains why the Rossby waves signal is not a pure signal on Figure 1 and 2. Using the RT approach method enhances our confidence in the presence of Rossby waves since we are able to detect them without the need for specific subjective filters."**

Figure 4 shows a relatively flat 2D-RT phase spectrum for angles less than 45 deg. Is there any evidence the purported peak at 21 deg is actually statistically significant over that background noise? Chelton recanted his concept of fast internal Rossby waves in the satellite observations by instead re-casting the preponderance of westward propagating signals as evidence of nonlinear eddies rather than waves. Only the largest scales seemed to be wavelike in behavior, but they were difficult to dissociate from the energetic mesoscale that dominated their analysis (Chelton et al., 2011). The authors in the present study give no indication that they are aware of or concerned about this updated framework.

**Following the Reviewer's comment, we made sure that the peak at 21 deg is significantly different than the background noise. We generated 50 random noise fields and added them to the 2D-RT before recomputing its energy curve. The random fields extend between -2 standard deviations and +2 standard deviations computed from the original 2D-RT. In 82% of**

the cases, the peak remained within 1° of 21°, while 98% were comprised within 2°.

Accordingly, we added the following text to the manuscript: "In order to make sure that the peak at 21° is significantly different than the background noise, we generated 50 random noise fields and added them to the 2D-RT before recomputing its energy curve. The random fields extend between -2 and +2 standard deviations computed from the original 2D-RT. In 82% of the cases, the peak remained within 1° of 21°, while 98% were comprised within 2°."

The eddies described in (Chelton et al., 2011) have an average lifetime of 32 weeks and an average propagation distance of 550 km, which are much smaller scales than the Rossby waves we describe. In addition, the wavelength of these Rossby waves (1066 km) is much larger than the Rossby radius of deformation (30 km) at the latitude of interest, indicating the observed westward propagation to be mostly due to waves rather than eddies. However, in order to differenciate the part of the westward signal from the 2D-RT that is due to Rossby waves against possible eddies, we propose the following future experiments. 1) Performing a tracking of eddies for different latitudes based on the DRAKKAR SSH fields. 2) From DRAKKAR temperature and salinity fields, calculate the densities, the baroclinic eigen modes, the steric height, and the eddie propagation relative to each baroclinic mode. This would also help future studies to analyse how the NAO signal is carried to the westward propagating SSH, and how this SSH variability impacts in turn the GS.

We thus added the reference proposed by the Reviewer #2 and the following text in the manuscript: "Using satellite observations, Chelton et al. (2011) showed that a part of the westward-propagating SSH variability is due to nonlinear mesoscale eddies. However, the eddies they found have an average lifetime of 32 weeks and an average propagation distance of 550 km, which are much smaller scales than the Rossby waves we describe. In addition, the wavelength of these Rossby waves (1066 km) is much larger than the Rossby radius of deformation (30 km) at 39° N, indicating the observed westward propagation to be mostly due to waves rather than eddies. In order to further quantify the part of the westward signal from the 2D-RT that is due to Rossby waves against possible eddies, we propose the following future experiments. 1) Performing a tracking of eddies for different latitudes based on the DRAKKAR SSH fields. 2) From DRAKKAR temperature and salinity fields, calculate the densities, the baroclinic eigen modes, the steric height, and the eddie propagation relative to each baroclinic mode. This would also help future studies to analyse how the NAO signal is carried to the westward propagating SSH, and how this SSH variability impacts in turn the GS."

The SSH signals include components from multiple vertical modes, which can obscure the signal they are after. Since they have a model, it would be more far more interesting to analyze the full model fields rather than just looking at SSH. Separate the flows into vertical modes, including the barotropic and 1st baroclinic, and maybe even the 2nd baroclinic, so that you can be more unambiguous in following only one internal mode, presumably the 1st baroclinic.

We agree with the Reviewer #2 that working with the 3D fields of the model outputs would be interesting in view of projecting them on the

**different modes of propagation, see our previous answer. We added this idea in the perspectives for future research, but this analysis would be outside of the scope of this work.**

It's not just the NAO forcing that drives the waves. It is the zonally and temporally integrated upstream effects of the wind-stress curl for all time scales at each longitude that matters locally in the oceanic response. Hence, showing lagged correlations of indices in Figure 7 is not particularly interesting or illuminating.

**We addressed this issue in the manuscript: "Considering the NAO signal transfers momentum through the wind stress to the central part of the NA (Visbeck et al., 1998), with a maximum impact on the SSH in this area (Esselborn and Eden, 2001), we use hereafter the longitude 30°W as a reference for the perturbation initiating a westward SSH Rossby wave."**

Finally, Figure 8 looks like two (essentially) uncorrelated red-noise time series.

**We agree that the Fig 8 does not show statistically significant correlations, which is described in the manuscript. We also think that these figures are nevertheless interesting. These weak correlations, together with the analysis in Fig. 9, are an argument to encourage further research in the physical processes linking NAO and Rossby waves.**

**We adapted the text by adding "While the correlations are not significant, the figures are nevertheless interesting. These weak correlations are an argument to encourage further research in the physical processes linking NAO and Rossby waves."**

Minor points:

1) All the figure captions should indicate that "model SSH" is being analyzed and plotted.

**We adapted all figures accordingly.**

2) Throughout the text, many of the numerical estimates indicate "four significant digits" accuracy. Really?

**We have limited the estimates to less digits in several places in the manuscript.**

Reference: Chelton, et al., 2011: Global observations of nonlinear mesoscale eddies. Prog. Oceanogr., 91, 167-2

---

## Editor Comment (EC1) · Andrew Moore (Editor) · 18 Dec 2020

Dear Author,

Based on the original comments of the two reviewers, and your responses, I am afraid that I cannot encourage you to submit a revised manuscript.

Both reviewers express the opinion that while you present some interesting results there is not enough "new" or "convincing" to warrant publication. More analysis and very major revisions would be required. Therefore, while I appreciate your responses to the detailed reviewer comments, the manuscript revisions that you propose fall short

of what would be required for the manuscript to be publishable.

Thank you for your submission to Ocean Science, but I regret to inform you that I must reject your manuscript at this time.

Regards Andrew Moore